# A Quadrivalent mRNA Immunization Elicits Potent Immune Responses against Multiple Orthopoxviral Antigens and Neutralization of Monkeypox Virus in Rodent Models

**DOI:** 10.3390/vaccines12040385

**Published:** 2024-04-05

**Authors:** Caixia Su, Sha Li, Yang Wen, Xiya Geng, Quanyi Yin, Yun Wang, Yelin Xiong, Zhihua Liu

**Affiliations:** 1Department of Research and Development, Yither Biotech Co., Ltd., Pudong, Shanghai 200120, China; 2State Key Laboratory of Virology, Center for Biosafety Mega-Science, Wuhan Institute of Virology, Chinese Academy of Sciences, 44 Hongshancelu Avenue, Wuhan 430071, China; lisha@wh.iov.cn (S.L.); wangyun@wh.iov.cn (Y.W.); 3University of Chinese Academy of Sciences, No. 19(A) Yuquan Road, Shijingshan District, Beijing 100049, China; 4Ab&B Biotech Co., Ltd., Taizhou 225300, China

**Keywords:** orthopoxvirus, vaccinia, MPXV, mRNA vaccine, cross immunogenicity

## Abstract

The global outbreak of the 2022 monkeypox virus infection of humans and the 2023 documentation of a more virulent monkeypox in the Democratic Republic of the Congo raised public health concerns about the threat of human-to-human transmission of zoonotic diseases. Currently available vaccines may not be sufficient to contain outbreaks of a more transmissible and pathogenic orthopoxvirus. Development of a safe, effective, and scalable vaccine against orthopoxviruses to stockpile for future emergencies is imminent. In this study, we have developed an mRNA vaccine candidate, ALAB-LNP, expressing four vaccinia viral antigens A27, L1, A33, and B5 in tandem in one molecule, and evaluated the vaccine immunogenicity in rodent models. Immunization of animals with the candidate mRNA vaccine induced a potent cellular immune response and long-lasting antigen-specific binding antibody and neutralizing antibody responses against vaccinia virus. Strikingly, the sera from the vaccine-immunized mice cross-reacted with all four homologous antigens of multiple orthopoxviruses and neutralized monkeypox virus in vitro, holding promise for this mRNA vaccine candidate to be used for protection of humans from the infection of monkeypox and other orthopoxvirus.

## 1. Introduction

Orthopoxviruses belong to the family Poxviridae and the genus orthopoxvirus contains 12 officially recognized species by the international committee on taxonomy of viruses, including Abatino macacapox virus, Akhmeta virus, Camelpox virus (CMLV), Cowpox virus (CPXV), Ectromelia virus, Monkeypox virus (MPXV), Raccoonpox virus, Skunkpox virus, Taterapox virus, Vaccinia virus (VACV), Variola virus (VARV), and Volepox virus, with Alaska poxvirus being the latest, informally classified member. Smallpox (caused by variola virus) was one of the most devastating human diseases that caused millions of deaths before it was eradicated in 1980 [1]. Following the eradication of smallpox, the risk of variola infection has been minimal and the vaccination program of smallpox was discontinued. It was only in 2022, when monkeypox broke out globally [2,3,4], that the concern of an epidemic of a human zoonotic orthopoxvirus was brought back to the public and health authorities. The 2022 global epidemic of monkeypox with confirmed cases of over 80 thousand was caused by the clade 2b monkeypox virus (a newly classified lineage of clade 2) [3]. In the following year (2023), a sexually transmissible and more virulent clade I of the monkeypox virus was documented in the Democratic Republic of Congo [5]. It is concerning that the recurrence of the monkeypox outbreak remains, due to the waning global immunity against smallpox caused by the discontinuation of the smallpox vaccination program [6]. The immunosuppression caused by ongoing HIV prevalence and other factors could lead to greater vulnerability to monkeypox resulting in high risks of severe disease and death [6]. The continuing monkeypox epidemic in African countries could result in accelerated viral evolution and adaptation to human-to-human transmission of the zoonotic disease [6]. All of the above brought much attention to the control of the monkeypox epidemic, such that WHO recommended the development of scalable less-reactogenic vaccines (such as mRNA vaccines) to improve efficacy and durability of protection for the control of an outbreak [6].

In addition to the concern of the global epidemic of monkeypox, there are threats of other zoonotic orthopoxviruses to human health. Cowpox infection of humans has occurred, with the host being domestic cats or elephants [7,8]. Camelpox virus [9] and Akhmeta virus [10] were also reported to be pathogenic to humans. The most recently described orthopoxvirus species that infected humans is Alaska poxvirus, first isolated in 2015 [11]. Taking into consideration the above, the development of a vaccine with broad protection against orthopoxviruses is desirable.

Extensively used smallpox vaccines are vaccinia virus-based. The first-generation vaccine, e.g., Dryvax, was produced in live animals and was effectively used for smallpox eradication. However, the use of the first-generation vaccine for mass vaccination is currently limited due to potentially severe complications (such as skin lesions, myocarditis, and neurovirulence) and manufactural contamination with bacteria and bacterial debris [12,13]. The second-generation vaccines are further attenuated vaccinia virus, ACAM2000, produced in cell lines [14]. Although cell culture-based vaccines are produced according to modern standards, second-generation smallpox vaccines can also cause serious adverse reactions during vaccination [15,16]. In addition to these safety issues, there is a limited reserve of first and second-generation vaccines [6]. The third-generation vaccines were further attenuated (replication-defective in mammalian cells) vaccinia virus (Ankara) vaccine MVA-BN [16,17] and a similar minimally-replicating LC16m8 (with restricted use in Japan) [18]. MVA-BN has passed clinical trials, including studies in patients with atopic dermatitis and HIV [19,20,21,22]. This two-dose live non-replicating vaccine MVA-BN was approved in the EU, Canada, and the USA, originally for smallpox immunization and later expanded to be used against monkeypox [23]. However, the vaccine efficacy of MVA-BN against monkeypox in at-risk populations needs to be assessed during cohort studies since a two-shot MVA-BN immunization in non-primed individuals yielded relatively low levels of MPXV-neutralizing antibodies [24]. Additionally, the vaccination coverage by the third generation even in the United States remains low due to constrained vaccine supply [5]. Therefore, the development of a new generation of vaccines with higher efficacy and higher productivity as well as better safety profiles is in urgent demand to meet the global vaccine requirement, in case of an epidemic of monkeypox or other pathogenic orthopoxviruses.

Through the use of a new set of bioinformatic stools to annotate the gene sets of representative strains of all species in the genus orthopoxvirus, it demonstrates that all existing orthopoxvirus species contain genes that are present in all members of the species, and in particular, the central core region of the genome encodes genes that are involved in basic virus replicative processes are much more conserved [25]. Experimental evidence from animal studies reveals that infection with any one orthopoxvirus produces substantial protection against disease produced by any other orthopoxvirus [26]. This was the foundation for the vaccinia virus being an effective vaccine against smallpox. Based on the above, we reasonably speculate that a vaccine targeting proper vaccinia viral antigens may provide protective immune responses against broad orthopoxviruses.

The vaccinia virus, like other orthopoxviruses, contains a linear double-stranded DNA genome [2], encoding about 200 proteins, including various proteins and enzymes required for virus replication, virus assembly, host restriction, pathogenicity, and other processes. The virus often exists in two different infectious forms, the intracellular mature viruses (IMV) and extracellular enveloped viruses (EEV), whose surface glycoproteins infect cells using different mechanisms [27]. Early studies show that the four membrane proteins A27, L1, A33, and B5 of the vaccinia virus are involved in the adsorption, binding, and intercellular transmission of virus-infected cells [28,29,30]. The A27 protein and the L1 protein on IMVs are generally considered to mediate the attachment and binding of the virus to cells, while the A33 protein and B5 protein on EEVs are considered to mediate the spread of the virus between cells. Polyclonal and monoclonal antibodies against these four proteins have a high level of neutralizing activity to the vaccinia virus [31,32,33,34,35,36], monkeypox virus [35], cowpox virus [35], and variola virus [35]. Recombinant protein platform and DNA platform used to deliver A27, L1, A33, and B5 alone or in combination can induce protective neutralizing antibody responses in mice [37,38,39,40] and monkeys [41] against vaccinia virus and monkeypox virus. Sequence analysis confirmed that these four vaccinia viral antigens and their homologous proteins in the monkeypox virus (A29, M1, A35, and B6, respectively) and smallpox virus are highly conserved (the conservation scores of all antigens are ≥93%, among which L1 is the most conserved, with a conservation ≥ 98.8%) [38]. Taking together, a vaccine targeting vaccinia viral antigens A27, L1, A33, and B5 may provide protective immune responses against broad orthopoxviruses including the monkeypox virus.

Recently, mRNA vaccines have attracted much more attention due to their excellent immunogenicity, short preparation time, high yield, and good safety data, which have been verified by intensive use of the two marketed COVID-19 mRNA vaccines. In this study, a novel poxvirus mRNA vaccine candidate, ALAB-LNP (lipid nanoparticle) was designed to encode four vaccinia viral antigens A27, L1, A33, and B5 tandemly in one molecule in comparison to a mixture of four mRNAs, each expressing a single gene of A27, L1, A33, or B5, to form a vaccine, 4Sin-LNP. Strong humoral and cellular immune responses were induced by both ALAB-LNP (mouse and rat model) and 4Sin-LNP (mouse model). However, ALAB-LNP elicited significantly stronger neutralizing antibody responses against both vaccinia virus and monkeypox virus in a dose-dependent manner (mouse model) in comparison to 4Sin-LNP. Furthermore, the neutralizing antibody induced by ALAB-LNP can last at least 6 months both in mice and rats. ALAB-LNP-induced cross-reactivity was demonstrated by equivalent or better serum IgG in responses to corresponding antigens from multiple orthopoxviruses (including cowpox, camelpox, variola, and monkeypox), and cross-neutralizing activity to the monkeypox virus. Together, these data suggest that the mRNA vaccine candidate ALAB-LNP has the potential to be an effective vaccine against infection of monkeypox, smallpox, and potentially other orthopoxviruses.

## 2. Materials and Methods

### 2.1. Cell and Virus

Hela S3 and VERO E6 cell lines were purchased from Procell Life Science & Technology Co., Ltd. (Wuhan, China) and cultured in DMEM medium (Gibco, Grand Island, NY, USA, 11965118) supplemented with 10% FBS (Gibco, 10091148) and 1% penicillin and streptomycin (Gibco, 15070-063), respectively. Vaccinia virus (ATCC-VR-1354) was purchased from ATCC (Manassas, VA, USA) and grown in Hela S3 and VERO E6. The virus titer was determined by VERO E6-based plaque assay. Monkeypox virus (IVCAS 6.9141, Clade IIb) was isolated from a patient in Wuhan (China) and propagated in Vero E6 cells prior to MPXV neutralizing assay.

### 2.2. Animal Study

6–8-week-old female BALB/c mice and 6–8-week-old female Sprague Dawley (SD) rats used for the experiments were grown under specific pathogen-free conditions at PharmaLegacy Laboratories (PLL, Shanghai, China) Co., Ltd. or Wuhan institute of virology (Chinese academic of sciences). To evaluate the immunogenicity of ALAB-LNP and 4Sin-LNP in mice, animals were randomly divided into seven groups: (1) group given empty-LNP, (2) group given 5 μg ALAB-LNP, (3) group given 10 μg ALAB-LNP, (4) group given 20 μg ALAB-LNP, (5) group given 5 μg 4Sin-LNP, (6) group given 10 μg 4Sin-LNP, and (7) group given 20 μg 4Sin-LNP. Rats were divided into four groups: (1) group given empty-LNP, (2) group given 30 μg ALAB-LNP, (3) group given 60 μg ALAB-LNP, and (4) group given 100 μg ALAB-LNP. Animals were immunized twice at an interval of 4 weeks with mRNA vaccine candidate ALAB-LNP, 4Sin-LNP, or the same volume of empty-LNP. Sera of immunized animals were collected for antigen-specific binding antibody detection, neutralizing antibody assay against VACV or MPXV, and heterologous antigen cross-reactive antibody examination. With regard to the dose-ranging study in mice, sera were collected on day 21 and day 42 post-primary-immunization, while rats’ sera were collected on day 42 post-primary-immunization. With regard to the monkeypox virus neutralization assay, sera were collected on day 42 post-primary-immunization. In the immune durability study, sera were collected on day 0, day 21, day 42, day 84, day 112, day 140, day 168, and day 196 from three groups of mice given 5 μg ALAB-LNP, 10 μg ALAB-LNP, and 20 μg ALAB-LNP, respectively, and two groups of rats given 60 μg ALAB-LNP, and 100 μg ALAB-LNP, respectively, for detection of binding and neutralizing antibodies.

Spleens were harvested for evaluation of cellular immunity 28 days post the boost immunization. With regard to vaccinia Tian-tan strain (VTT) immunization, 6-week-old female BALB/c mice were intraperitoneally vaccinated with VTT of 1 × 10^4^, 1 × 10^5^, or 1 × 10^6^ plaque-forming units (pfu) twice at an interval of 3 weeks, followed by sera collection at 2 weeks post-boosting-immunization for analyzing of neutralizing antibody against MPXV.

### 2.3. Construction of Recombinant Plasmid

The mRNA of pox vaccine candidate ALAB-LNP was designed by insertion of VACV A27, L1, A33, and B5 genes into the coding region of the plasmid, pUCYH, in tandem with a modified P2A linker between each gene. A T7 promoter sequence and a 5′ UTR sequence including Kozak sequence were added to the upstream of the genes and a stop codon TGATAA, 3′ UTR, PolyA, and BspQ I cleavage sites were added to the downstream. The target genes were next cloned into pUCYH plasmid to create plasmid ALAB-pUCYH, which was confirmed by Sanger sequencing. To generate the 4Sin mRNAs, 4 plasmids were constructed by inserting A27, L1R, A33, or B5R genes respectively, using the same backbone of mRNA ALAB.

### 2.4. ALAB mRNA Preparation

The recombinant plasmids encoding ALAB and four single genes (Sin1 to Sin4) were extracted and linearized by restriction endonuclease BspQ I cleavage followed by plasmid recovery and purification. In vitro transcription reaction (IVT in 50 μL) was performed at 37 °C for 3 h after vortexing of the mixture containing linearized plasmid template, ATP (100 mM), GTP (100 mM), CTP (100 mM), UTP (100 mM), Cap analogue (100 mM, Vazyme, Nanjing, China), T7 RNA polymerase (200 U/μL, Vazyme), 10X T7 Reaction Solution (Vazyme), RNase enzyme inhibitor (40 U/μL, Vazyme), pyrophosphatase (0.1 U/μL, Vazyme), and RNase Free H_2_O (Invitrogen, Carlsbad, CA, USA, 10977015). After the IVT, 170 μL of Rnase-free H_2_O, 5 μL of DNaseI enzyme (NEB, Beverly, MA, USA, M0303L), and 25 μL of 10× DNaseI enzyme reaction solution (NEB, M0303L) were added to the mRNA and incubated at 37 °C for 20 min. Purified mRNA was obtained using OligoDT beads (Vazyme, N401). The mRNA integrity and purity were analyzed using Agarose Gel (Bio-Rad, Hercules, CA, USA) and HPLC-SEC (Thermo Fisher Scientific, Waltham, MA, USA).

### 2.5. mRNA-LNP Preparation

Purified ALAB mRNA was diluted to 167 μg/mL, and each of the four Sin mRNAs was diluted to 41.75 μg/mL, with 50 mM citric acid buffer, pH 4.0, to obtain an aqueous solution. The organic phase solution was prepared by dissolving ionizable lipid, DSPC, cholesterol, and PEG2000-lipid in ethanol. The aqueous and organic phase solutions were siphoned into the INano™ L injector (MicroNano, Shanghai, China) at a flow ratio of 3:1 with a total flow rate of 12 mL/min for mRNA encapsulation. The mRNA-LNP complex was next rapidly diluted in PBS (pH 7.4) at room temperature and centrifuged at 3000 rpm at 4 °C for 10 min in a 100 kD ultrafiltration tube (Merck Millipore, Billerica, MA, USA). The mRNA-LNP complex was concentrated to 0.4 mg/mL followed by filtration using a 0.22 μm filter to sterilize the ALAB-LNP and 4Sin-LNP vaccine candidates. Particle sizes and polymer dispersity index (PDI) of the mRNA vaccines were analyzed using a Malvern particle size analyzer. The encapsulation rates of the mRNA were determined by RiboGreen (Invitrogen™, R11490) nucleic acid assay, with the empty-LNP being the negative control.

### 2.6. Detection of Antigen Expression Using Flow Cytometry

Briefly, 2 × 10^6^ of HEK293T cells were transfected with 5 μg ALAB mRNA or 5 μg 4Sin mRNA using TransIT™-mRNA Transfection Kit (Mirus Biotech, Madison, WI, USA, MIR-2250) or with transfection Kit alone. After 24 h culturing, cells were detached from the six-well plate surface with PBS containing 3% FBS, followed by staining with aqua fluorescent reactive dye (Invitrogen, L34966A, 1:1000) to differentiate alive versa dead. Cells were fixed with IC Fixation Buffer (Invitrogen, 00-8222-49), permeabilized (Invitrogen, 008333-56) and blocked with CD16/CD32 monoclonal antibody (Invitrogen, 14-0161-85, 1:1000). Finally, the cells were stained with monoclonal antibodies (Henxy Biotech, Shanghai, China, RAB-2412, RAB-2422, RAB-2471, RAB-2442) that specifically recognizes A27, L1, A33, or B5 (1:200 dilution), respectively, followed by adding anti-mouse AF488 (Invitrogen, A55058, 1:500) or anti-human AF488 (Invitrogen, A-11013 1:500) as secondary antibodies. Cells were acquired on a FACS Attune NxT Acoustic Focusing Cytometer (Thermo Fisher Scientific, Waltham, MA, USA) and data were analyzed using FlowJo 10.8.1.

### 2.7. Antigen-Specific Binding Antibody Detection Assay

ELISA was used to measure the titer of antigen-specific antibody IgG in mouse or rat sera. Each antigen (A27, A33, L1, and B5) was diluted to 1 μg/mL with coating buffer followed by coating the 96-well plates with each antigen, respectively, at 4 °C overnight. The serum was serially diluted by 3-fold and placed into the plate coated with 5% milk at 37 °C for 1 h. After washing with PBST, the plate was incubated with HRP-conjugated goat anti-mouse IgG (Southern Biotech, Birmingham, AL, USA, 1031-05, 1:4000,) or HRP-conjugated goat anti-rat IgG (Abcam, Cambridge, MA, USA, ab205720, 1:4000) at 37 °C for 1 h and color reaction was developed by adding 100 μL TMB (Invitrogen™, 002023) for 5 min followed by adding 50 μL of ELISA terminating solution (New Cell&Molecular Biotech, Suzhou, China, E40500) to stop the reaction. The absorbance was measured at 450/620 nm using an ELISA plate reader (Thermo Fisher Scientific, Waltham, MA, USA), Varioskan LUX). The mean OD (optical density) of sera from naïve mice was multiplied by 2.1 to define the positive cutoff point. The cross-binding activity of sera (collected at 14 days post-boosting) from mice immunized with 20 μg ALAB-LNP against homologous antigens of A27, L1, A33, and B5 of vaccinia, monkeypox, camelpox, cowpox, and variola virus was detected, respectively, by ELISA as described above. In this study, the limit of detection for the titer of antigen-specific binding antibodies against multiple antigens was 200. Sera from unimmunized animals were set as negative controls for this assay.

### 2.8. Neutralizing Assay

Sera from immunized mice were inactivated at 56 °C for 30 min, followed by 5 serial dilutions (3-fold) with serum-free DMEM medium (Gibco, Grand Island, NY, USA) with or without 20% baby rabbit complement (Cedarlane Laboratories, Hornby, Ontario, Canada, CL3441-S50-R), starting from 1: 50. 100 μL of serum was next mixed with an equal volume of VACV (300 pfu) and incubated at 37 °C for 1 h. The mixture was added to the 48-well cell plates laid with VERO E6 cells one day prior (1.5 × 10^5^ cells/well) and incubated for 2 h before replacing media with DMEM containing 2% FBS (*v*/*v*) and 0.3% (*w*/*v*) methylcellulose (Sinopharm, Shanghai, China, LA1810501). After incubation for 28 h, the cells were fixed with 4% PFA, stained with crystal violet solution for 0.5 h, and rinsed with distilled water. Finally, the plates were scanned and plaques were counted using an enzyme-linked immune-spot analyzer (Cellular Technology Ltd., Shaker Heights, USA, S6 Universal). In the NT_50_ calculation procedure, the infection inhibition rates of each dilution of the sample are calculated according to the plaque number as follows: the inhibition rate = [1 − (average plaque number of sample − average plaque number of cell control)/(average plaque number of virus control − average plaque number of cell control)] × 100%

The NT_50_ is calculated following the Reed and Muench method on the basis of the results of the inhibition rate.

The detection of neutralizing antibodies against monkeypox virus (IVCAS 6.9141) was similar to that against VACV. The inactivated and diluted serum (collected at 2 weeks post-boosting for ALAB-LNP, 4Sin-LNP and VTT groups) was incubated with the MPXV (50 pfu) for 1 h, followed by adding to the VERO E6 cells placed in 24-well plate one day prior (2 × 10^5^ cells/well) and incubated for 2 h before replacing the media with DMEM containing 2% FBS and 0.3% (*w*/*v*) methylcellulose (Sinopharm, LA1810501). After 4 days, plaques were counted by crystal violet staining, and neutralizing antibody titers were calculated as described above for VACV neutralization.

To determine the role of anti-L1 antibody in serum neutralizing activity against VACV, three concentrations (0.2 μg, 2 μg, 20 μg) of VACV L1 protein and its homologous protein M1 of MPXV were incubated with inactivated mouse serum, respectively, at 4 °C overnight. Neutralizing antibody titers were measured as described above and the inhibition rates of VACV by the serum were calculated according to the Reed and Muench method. In this study, the limit of detection for the titer of neutralizing antibody against Vaccinia virus or monkeypox virus was 50, unless otherwise specified. In neutralization assays, cells infected with the virus in the presence of sera from non-immune animals were set as negative controls for neutralization, and cells infected with virus only (without adding sera) were set as positive control of infection.

### 2.9. ELISPOT Assay for Cellular Immune Response Detection

The ELISPOT plate (Millipore, MSIPS4W10) was filled with 35% ethanol at 50 μL/well for 1 min. The liquid was discarded and the plate was washed with sterile deionized water at 200 μL/well 5 times. IFN-γ capturing antibody (MABTECH, Nacka, Sweden, 3321-2H) was diluted to 15 μg/mL with PBS and added to the above plates at 100 μL/well, and the plates were incubated at 4 °C overnight. On the next day, the plates were washed with PBS, and RPMI 1640 medium (containing 10% FBS) was added followed by incubation at room temperature for 30 min. Single-cell suspension was obtained from a mouse spleen and cell numbers were counted after lysis of erythrocyte. The culture medium in the plate was discarded and appropriate numbers of cells were added. Next, peptides (Peptides pool of A27, L1-1, L1-2, A33, B5-1, and B5-2) were added at a final concentration of 2.5 μg/mL each. The plates were placed in an incubator set at 37 °C for 36 h followed by discarding the cells and washing with PBS. The diluted detection antibody (MABTECH, 3321-2H, 1:1000) was added and the plate was incubated at room temperature for 2 h. The diluted streptavidin-HRP (MABTECH, 3321-2H, 1:1000) was added and the plate was incubated at room temperature for 1 h. Finally, TMB solution (MABTECH, 3651-10) was added followed by washing with deionized water after spots became obvious. After the plates were dried at room temperature, pictures were taken and the spots in the wells were counted using an enzyme-linked immune-spot analyzer (Cellular Technology Ltd., Shaker Heights, USA, S6 Universal).

### 2.10. Safety Evaluation of ALAB-LNP Vaccine in Rodent Model

Mice or rats were immunized twice following an interval of 4 weeks, respectively. Animal survival and side reactions were monitored and recorded every two days from day 0 to day 42 post-primary-immunization. The clinical scoring criteria for side reactions of the injection site are as follows: 1 represents dropsy, 2 represents swelling, 3 represents stiffness, and 4 represents immobility.

### 2.11. Statistical Analysis

Statistical analysis was performed using GraphPad Prism 8.0 (GraphPad Software, San Diego, CA, USA) and results are presented as mean ± standard error of the mean (SEM). Spearman correlation analysis of neutralizing antibody titers against VACV and MPXV was conducted using SPSS 22 to analyze the results of 30 samples, and *p* values < 0.05 were considered to be statistically significant. The exact sample size (n) for each experimental group is indicated in the figure legends. Differences between the two groups were analyzed with unpaired *t*-tests or two-way analysis of variance (ANOVA). Significance level was marked in figures and annotated in the figure legends.

## 3. Results

### 3.1. Design, Preparation, and Characterization of the mRNA Candidates

A conventional mRNA structure contains a cap structure, a 5’UTR untranslated region, a CDS translated region, a 3’UTR untranslated region, and a PolyA structure. Four VACV genes, A27, L1, A33, and B5 separated by modified P2A linkers, were inserted into the CDS translation region (Figure 1A) to result in the vaccine candidate mRNA, ALAB. Four mRNAs encoding A27, A33, B5, and L1 (sin1, sin2, sin3, and sin4) were also constructed, respectively, using the same mRNA backbone (Figure 1A). Mixing of equal amounts of sin1, sin2, sin3, and sin4 resulted in the second vaccine candidate mRNA, 4Sin (Figure 1A).

VACV genes were first cloned into plasmid pUCYH followed by pDNA extraction, linearization, purification, and in vitro transcription to obtain ALAB and sin1, sin2, sin3, and sin4 mRNA molecules, respectively. Integrity and purity of the synthesized mRNA were demonstrated using agarose gel electrophoresis and HPLC-SEC analysis. The capping efficiency of each molecule was higher than 90%.

The transfection efficiency was demonstrated by flow cytometry that the expression rates of A27, L1, A33, and B5 were 37.1%, 83.6%, 87.4%, and 83.8%, respectively, after cells were transfected with ALAB mRNA (Figure 1B), while the expression rates of mixture of the four genes were 67.3%, 40.1%, 96.3%, and 91.6% (Figure 1B), respectively, after cells were transfected with 4Sin mRNAs. The high levels of protein expression data predicted a promising potency of the mRNA candidates.

To generate the vaccine candidates, both ALAB mRNA and 4Sin mRNAs were fully mixed with lipids using microfluidic technology to form ALAB-LNP and 4Sin-LNP, respectively. The particle sizes of ALAB-LNP, 4Sin-LNP, and empty-LNP were 80, 83, and 76 nanometers, respectively (Table 1). The PDI of ALAB-LNP, 4Sin-LNP, and empty-LNP were all less than 0.1 (Table 1), indicating uniformity of the nanoparticles. The encapsulation efficiencies (EE) of ALAB-LNP and 4Sin-LNP were 90% and 89% (Table 1), respectively.

### 3.2. mRNA Vaccine Candidates Induced Robust Antibody Responses and Cellular Immune Responses in Mice

To investigate the immunogenicity of ALAB-LNP and 4Sin-LNP in mice, we set a two-dose immunization regimen with a 4-week interval (Figure 2A). Among the groups of mice immunized with different amounts of mRNA of ALAB-LNP or 4Sin-LNP, significant levels of anti-L1-, anti-A33- and anti-B5-specific antibody responses were induced after the first immunization (with mean titers at 3–5 log, Figure 2C–E) with the exception of the anti-A27-specific antibody at the mean titers less than 3 log (Figure 2B). The antibody levels against the four antigens increased significantly after the boost immunization, with mean antibody titers of anti-L1, anti-A33, and anti-B5 reaching 5–7 log, and anti-A27, reaching 4–6 log (Figure 2B–E). When comparing the responses induced by ALAB-LNP and 4Sin-LNP, higher IgG titers of L1 (at 5, 10, or 20 μg per dose) or equivalent IgG titers of A33 (at 5 or 20 μg per dose), B5 (at 5, 10 or 20 μg per dose) and A27 (at 5 or 10 μg per dose) were found in mice immunized with ALAB-LNP in comparison to that with 4Sin-LNP (Figure 2B–E). Dose-dependent responses were clearly seen in anti-A27 IgG, anti-L1 IgG, and anti-A33 IgG in mice after the boost immunization with ALAB-LNP (Figure 2B–D), while in anti-A27 IgG and anti-L1 IgG in mice boosted with 4Sin-LNP (Figure 2B,C).

To explore whether the mRNA vaccine can induce specific cellular immune responses, the mouse splenocytes were stimulated with 6 antigen peptide libraries A27, L1-1, L1-2, A33, B5-1, and B5-2, respectively, for detection of IFN-γ produced by specific T cells using ELISPOT assay. Significant specific IFN-γ responses were detected to all four antigens (Figure 2F) with the highest being the responses to the A33 peptide pool (Figure 2F). The overall T cell response induced by ALAB-LNP and 4Sin-LNP was comparable at all three doses. Importantly, no cellular immunity was detected against the linker (Figure 2F).

### 3.3. mRNA Vaccine Candidates ALAB-LNP Induced Potent Neutralizing Antibody against VACV

In order to determine whether the antibodies induced by the ALAB-LNP and 4Sin-LNP have virus neutralization activity against VACV, we incubated the inactivated mice sera with VACV virus for detection of neutralizing activity that was calculated by plaque reduction. At 3 weeks post the first immunization, sera from 4Sin-LNP immunized mice did not exhibit neutralizing activity above the limit of detection (Figure 3A). On the contrary, the immune sera from ALAB-LNP vaccinated mice (at 5, 10, or 20 μg mRNA/dose, respectively), showed obvious neutralizing activity. After boosting, the neutralizing antibody titers of the mice given ALAB-LNP increased by 1.5–2 log and that of mice immunized with 4Sin-LNP increased by 1–2 log (Figure 3A). Strikingly, the neutralizing antibody titers (under the same immunization dose) in sera from mice immunized with ALAB-LNP were approximately 1 log higher than that from mice immunized with 4Sin-LNP (Figure 3A). Knowing from early data that the anti-L1 binding antibody in mice given ALAB-LNP was significantly higher than that given 4Sin-LNP (Figure 2C) prompted us to study if L1-specific antibody was the main contributor to the neutralizing activity against VACV. In the next study, VACV L1 and equivalent MPXV M1 proteins were incubated, respectively, with the sera from mice immunized with ALAB-LNP prior to neutralization of the virus. Both VACV L1 and MPXV M1 proteins could inhibit the neutralizing activity in a dose-dependent manner (at 0.2 μg/mL to 2 μg/mL, Figure 3B). At 20 μg/mL, the neutralizing activity against VACV was >90% inhibited by both VACV L1 and MPXV M1 proteins, suggesting that the neutralizing activity induced by ALAB-LNP was mainly L1 (M1) specific, and neutralizing antibody response against VACV (Figure 3A) correlated with the anti-L1 IgG responses (Figure 2C). Taken together, mRNA vaccine candidate ALAB-LNP induced a significantly stronger neutralizing response against VACV and the neutralization was mainly L1-specific.

### 3.4. mRNA Vaccine Candidates ALAB-LNP Induced Potent Neutralizing Antibody against MPXV

The neutralizing activity induced by ALAB-LNP and 4Sin-LNP to monkeypox virus was studied next. As shown, both ALAB-LNP and 4Sin-LNP induced significant levels of neutralizing antibodies at 10 μg/dose and 20 μg/dose, respectively (Figure 4A). The neutralizing antibody titers in mice given 5 μg and 20 μg of ALAB-LNP mRNA per dose, respectively, were significantly higher than that in mice given a similar dose of 4Sin-LNP, a similar neutralizing pattern seen in neutralization against VACV (Figure 3A). Among groups immunized with various amounts of ALAB-LNP mRNA, a dose-dependent trend of neutralization was found for MPXV (Figure 4A). The above data suggested that ALAB-LNP was a better candidate for MPXV than 4Sin-LNP. Parallel investigation on live attenuated smallpox vaccine, vaccinia Tian-tan strain (VTT), showed poor MPXV neutralizing activity of the vaccine even at 10^6^ pfu/dose while the neutralizing titer of ALAB-LNP was >3 log (Figure 4B), suggesting ALAB-LNP as a more potent vaccine than live-attenuated vaccinia Tian-tan strain.

Another important finding was that analysis of the neutralization activity against MPXV and VACV in this study showed a strong correlation between these two neutralization assays (Figure 4C), indicating that VACV neutralization assay may replace MPXV neutralization assay in immunology study and vaccine product quality characterization. Collectively, these data suggested that ALAB-LNP is a more efficacious vaccine candidate (than 4Sin-LNP) for further investigation.

### 3.5. ALAB-LNP Was Immunogenic in Rats

The immunogenicity of ALAB-LNP in rats was next confirmed by vaccination of SD rats with 30 μg, 60 μg, or 100 μg per dose, respectively, twice at a 4-week interval. At 2 weeks post-boosting-immunization, strong specific binding antibodies to 4 antigens were induced (Figure 5A), with the titers ranging from 4 to 6 log at various dosages. The neutralization assay showed robust neutralizing antibody responses against VACV without significant differences among the doses tested (Figure 5B). These data demonstrated the immunogenicity of the mRNA vaccine candidate in rats and indicated the SD rat as a relevant animal to support the safety evaluation of this vaccine candidate.

### 3.6. ALAB-LNP Induced Long-Lasting Immune Responses in Mice and Rats

The longevity of immune responses could determine the efficacy of a vaccine. To investigate the longevity of ALAB-LNP, mice or SD rats were immunized twice at a 4-week interval followed by testing of binding and neutralizing antibody responses at various points post-immunization, i.e., day 0, and 3, 6, 12, 16, 20, 24 and 28 weeks post the first immunization. In mice, serum IgG against 4 VACV antigens, A27, A33, L1, and B5 reached a peak at 6 weeks post-priming, and subsequently decreased by 1 to 2 logs, respectively, at 12 weeks post-immunization (Figure 6A–D). Interestingly, the antibody levels to all four antigens in mice given 10 μg or 20 μg per dose mRNA maintained from 12 to 28 weeks post-immunization so far (Figure 6A–D). Similar patterns of anti-A27 and anti-B5 were found in rats (Figure 6E,H), while antibody responses to A33 and L1 showed a delayed peak time (Figure 6F,G). The long-lasting immune responses were also seen with neutralizing antibodies. In mice, a high level of persistent neutralizing activity (NT_50_ titer > 10^3^) was detected from 6 weeks to 28 weeks post-prime (Figure 7A). In rats, the neutralizing activity peaked at 6 weeks post-prime (NT_50_ titer > 10^3^), slightly decreased in the following 6 weeks (NT_50_ titer > 10^2^), and remained stable till to 28 weeks post-prime immunization (Figure 7B). The longevity (one year and beyond) of the IgG responses and neutralizing activity is under investigation.

### 3.7. ALAB-LNP Induced Cross Binding Antibody Responses to Multiple Orthopoxviral Antigens

It is known that the four vaccinia virus antigens (A27, L1, A33, and B5) are highly conserved among monkeypox virus, cowpox virus, and smallpox virus. In this study, we tested whether the mRNA vaccine encoding these antigens could generate cross-immunity against homologous proteins of multiple orthopoxviruses (including VACV, VARV, CMLV, MPXV, and CPXV). Immune sera from mice immunized with ALAB-LNP were serially diluted and incubated with each of the homologous proteins, respectively, to detect cross-binding antibody titers. ELISA results showed that robust IgG responses were detected against four viral antigens of VARV, CMLV, MPXV, and CPXV, respectively, with various immunization dosages (Figure 8A–D).

### 3.8. ALAB-LNP Induced Complement-Dependent Enhancement of Neutralizing Activity

Early studies had demonstrated that the addition of complement to serological assay for VACV and MPXV could promote neutralizing activity [42]. To investigate if the above applied to the neutralization induced by ALAB-LNP, sera from ALAB-LNP immunized mice or rats were incubated, respectively, with or without 10% of one of the two types of complements, guinea pig complement (GP-C) and baby rabbit complement (BR-C), prior to mixing with VACV. Interestingly, both types of complement significantly boosted the neutralizing activity induced by ALAB-LNP in mice, i.e., a 2-fold increase induced by GP-C and 7-fold by BR-C, respectively (Figure 9A). In sera from rats, the enhancement of neutralization by complement was greater than that in mice, i.e., a 9-fold increase induced by GP-C and an 80-fold increase induced by BR-C (Figure 9B), suggesting both mice and rats are relevant animals for investigating the role of complement in neutralization of poxvirus.

### 3.9. Preliminary Safety Evaluation of ALAB-LNP in Mice and Rats

To evaluate the pre-clinical safety of the vaccine candidate, mice and rats immunized with ALAB-LNP were monitored for survival and side reactions including dropsy, swelling, stiffness, and immobility post-prime and boost-immunization. All mice and rats survived during the observation period from day 0 to day 42 post-primary-immunization (Figure 10A,C). After the primary immunization, rapid reactions were observed in both mice and rats on next day post-immunization and the reactions disappeared within 7 days in mice (Figure 10B). In rats, side reactions disappeared within 5 days post-prime-immunization (Figure 10D). Similar to priming, the boost immunization also caused a rapid onset of side reactions which diminished within 7 days for mice and 5 days for rats post-injection, respectively (Figure 10B,D). Of note, immobility was not found both in mice and rats during the course of the observation (Figure 10B,D), with the average highest reaction score being a short period of stiffness (approximately one day) followed by 3–4 days of swelling for mice and average highest reaction score being swelling (1–2 days) followed by dropsy for 2–3 days in the rat model. Overall, these data suggest a relatively safe mRNA vaccine candidate of ALAB-LNP in rodent models. A more stringent GLP study of pre-clinical safety of ALAB-LNP will be conducted in the future.

## 4. Discussion

Efforts have been made to develop orthopoxvirus vaccines, including live attenuated virus vaccine technology [14,16], replication-defective virus vaccine technology [17,18], DNA vaccine technology [40,43], and recombinant protein technology [41]. Each has its own limitations such as severe skin side effects of live attenuated vaccine [14], weak immunogenicity of replication-deficient virus vaccines [24] and DNA vaccines [43], and long development cycle of recombinant protein technology [44]. Encouraged by the two new COVID-19 mRNA vaccines, some researchers have also tried to use mRNA technology to develop monkeypox vaccines, through the designing of multiple mRNAs that express a single or fused monkeypox antigen [45,46,47,48,49,50]. Some of those designs elicit potent immunogenicity but will face challenges in delivering multiple mRNAs during process development and manufacturing. In this study, we designed a novel mRNA that delivered four vaccinia antigens in one molecule, ALAB-LNP, to elicit immune responses against infection by broad orthopoxviruses. Knowing that some vaccine candidates, targeting vaccinia viral proteins, A27, B5, L1, and A33 elicited neutralizing antibody responses against the vaccinia virus and monkeypox virus [38,40,41,51,52], we included the above four genes in our mRNA construct to induce potent immune responses. Meanwhile, earlier studies also showed that immunization with two DNA vaccines (encoding L1 and A33, respectively) coated on the same gold bead, did not induce a specific neutralizing antibody response that was detected. However, when the two DNA vaccines were coated on different gold beads, a stronger neutralizing response and greater protection were found than either immunogen alone [37], suggesting that co-delivery of the two genes in the same cell could induce immune suppression. To evaluate if the quadrivalent ALAB mRNA would generate discrepancy in immune responses compared to a mixture of four mRNAs (4Sin), each expressing one of the four antigens, A33, L1, A27, or B5, in vitro gene expression and in vivo immune responses were compared. Higher expression of L1 and equivalent expression of A33 and B5 were found after transfection of HEK 293 cells with ALAB mRNA in comparison to transfection with 4Sin mRNA (Figure 1B, with the exception of lower expression of A27). Vaccine candidate, ALAB-LNP, induced potent binding antibody immune responses (Figure 2B–E) with significantly higher anti-L1 IgG (at 5, 10, and 20 μg per dose) and equivalent anti-A33 IgG (at 5 and 20 μg per dose), anti-B5 IgG (at 5, 10 and 20 μg per dose) and anti-A27 IgG (at 5 μg and 10 μg per dose) in comparison to 4Sin-LNP (Figure 2B–E). Furthermore, equivalent total VACV-specific cellular IFN-γ response was detected between mice immunized with ALAB-LNP and those given similar doses of 4Sin-LNP (Figure 2F) although the cellular response specific to A33 seemed higher in mice immunized with 4Sin-LNP. Overall, the vaccine candidate ALAB-LNP induced a better or equivalent binding IgG response, and an equivalent IFN-γ T cell response, compared to 4Sin-LNP. Impressively, ALAB-LNP induced significantly stronger neutralizing antibody responses against both VACV (Figure 3A) and MPXV (Figure 4A), than 4Sin-LNP. More strikingly, the neutralizing antibody titer of the ALAB-LNP group was over 15-fold higher than the existing smallpox live-attenuated vaccine, Vaccinia Tian-tan strain (Figure 4B, given at 10^6^ pfu/dose), further indicating ALAB-LNP as a potent vaccine candidate. The significant difference in neutralization between ALAB-LNP and 4Sin-LNP could be due to the difference in expression levels of L1 induced by the two mRNAs, respectively (83.6% vs. 40.1%), suggesting a more potent expression of L1 induced by ALAB-LNP. During the animal studies, preliminary data showed that no obvious severe safety issues (e.g., death and immobility of animal) were observed after the mRNA vaccination in both the mouse and the rat models (Figure 10), indicating a safe mRNA vaccine candidate. The potential broad protection of mRNA ALAB-LNP was demonstrated by vaccine-induced cross-binding antibody activity to multiple orthopoxviral antigens and neutralizing activity to monkeypox in this study. The broad protection will be further validated in other animal studies and human trials.

In addition to the advantages of mRNA vaccines, other important features of this mRNA vaccine candidate ALAB-LNP include (1) a simpler manufacturing process than that for the mixture of multiple single mRNAs, i.e., manufacturing of one drug product instead of four, (2) better productivity than recombinant vaccine, and (3) reduced manufacturing time and (4) reduced cost, all of which making ALAB-LNP a more manufacturable vaccine candidate for potential pandemic of monkeypox virus and other pathogenic orthopoxviruses.

## 5. Conclusions

In summary, we have developed a novel quadrivalent mRNA pox virus vaccine candidate, ALAB-LNP, with sound safety, significantly potent immunogenicity, and potentially broad protecting capability against orthopoxviruses. The process development and manufacturing process are simplified, the processing time is shortened and production cost is reduced. All of the above hold promises for this mRNA vaccine candidate to be used for the prevention of a pandemic of monkeypox and other orthopoxviruses.

## 6. Patents

A patent related to this study has been applied by Yither Biotech and Ab&B Biotech (Publication number: CN116286913A).

## Figures and Tables

**Figure 1 vaccines-12-00385-f001:**
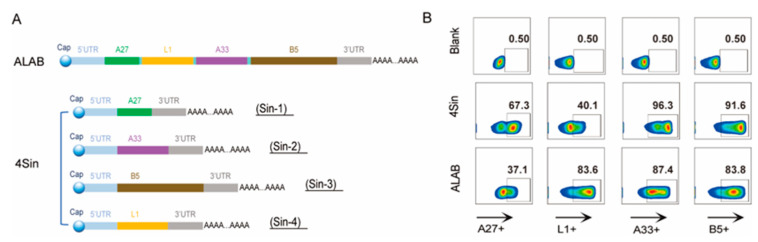
Novel design and characterization of mRNA used for generation of vaccine candidates. (**A**) Schematic diagram of the mRNA vaccine candidates used for vaccine formulation; Elements of mRNAs were showed in different color: 5′UTR (lightskyblue), 3′UTR (gray), A27 gene (green), L1 gene (glod), A33 gene (darkorchid), B5 gene (darkgoldenrod), modified P2A linker (cyan); (**B**) Characterization of antigen expression by flow cytometry.

**Figure 2 vaccines-12-00385-f002:**
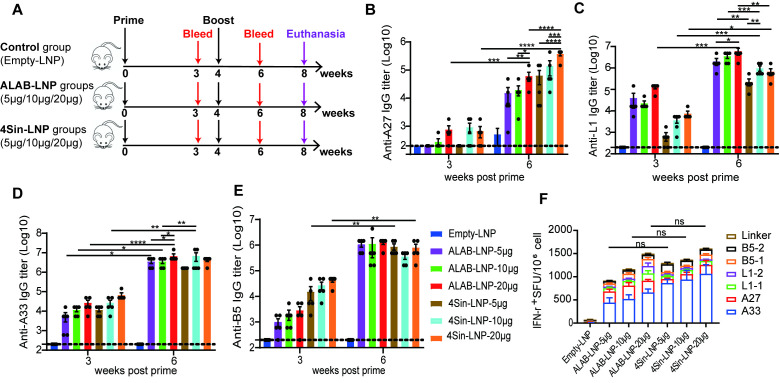
Immunogenicity of poxvirus mRNA vaccine candidates, ALAB-LNP, and 4Sin-LNP in mice. (**A**) Immunization schedule of mice (n = 5 for each group). The black, red, and purple arrows represent time points for immunizations, blood collection, and euthanasia; (**B**–**E**) Serum binding antibody titer against A27 (**B**), L1 (**C**), A33 (**D**) and B5 (**E**) at 3 or 6 weeks post-prime; (**F**) Cellular immune response in mice; Summary data are shown as means ± SEM; The dashed line indicates the limit of detection (LOD); Statistical significance was assessed by two-way ANOVA with Tukey’s or Sidak’s multiple comparisons test; *p* values were indicated by * (*p* < 0.05), ** (*p* < 0.01), *** (*p* < 0.001), or **** (*p* < 0.0001).

**Figure 3 vaccines-12-00385-f003:**
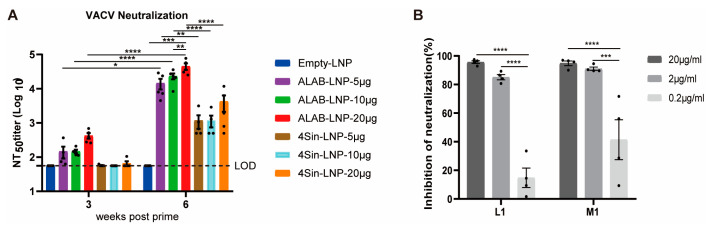
Neutralizing antibody against VACV induced by ALAB-LNP and 4Sin-LNP. (**A**) Serum VACV neutralization titer of mice immunized with ALAB-LNP or 4Sin-LNP at 3 or 6 weeks post-prime (n = 5 for each group); (**B**) Competitive inhibition of neutralization by VACV L1 protein and MPXV M1 protein; Summary data are shown as means  ±  SEM; The dashed line indicates the limit of detection (LOD); Statistical significance was assessed by two-way ANOVA with Tukey’s multiple comparisons test; *p* values were indicated by * (*p* < 0.05), ** (*p* < 0.01), *** (*p* < 0.001), or **** (*p* < 0.0001).

**Figure 4 vaccines-12-00385-f004:**
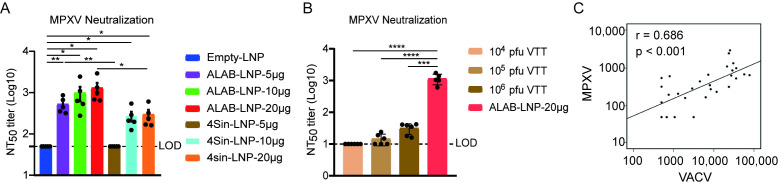
Cross-neutralizing activity against monkeypox virus. (**A**) Serum neutralizing activity of mice immunized with ALAB-LNP or 4Sin-LNP against monkeypox virus; (**B**) Significantly higher neutralizing activity induced by ALAB-LNP than smallpox live-attenuated vaccine, vaccinia Tian-tan strain (VTT); (**C**) Correlation of VACV neutralization assay and monkeypox neutralization assay (x-axis refers to VACV NT_50_ titer, y-axis refers to MPXV NT_50_ titer); Summary data are shown as means  ±  SEM; The dashed line indicates the limit of detection (LOD); Statistical significance was assessed by unpaired *t*-test; *p* values were indicated by * (*p* < 0.05), ** (*p* < 0.01), *** (*p* < 0.001), or **** (*p* < 0.0001). Correlation analysis was performed on SPSS.

**Figure 5 vaccines-12-00385-f005:**
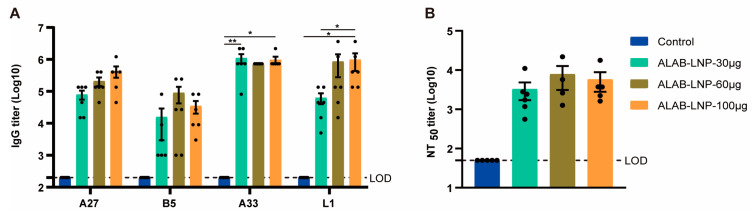
ALAB-LNP mRNA vaccine candidate induced immunity in rats. (**A**) Serum binding antibody titer against A27, L1, A33, and B5 at 6 weeks post-prime; (**B**) Serum neutralizing activity of rats immunized with ALAB-LNP against VACV. Summary data are shown as means ± SEM; The dashed line indicates the limit of detection (LOD); Statistical significance was assessed by two-way ANOVA with Tukey’s multiple comparisons test (**A**) or unpaired *t*-test (**B**); *p* values were indicated by * (*p* < 0.05), ** (*p* < 0.01).

**Figure 6 vaccines-12-00385-f006:**
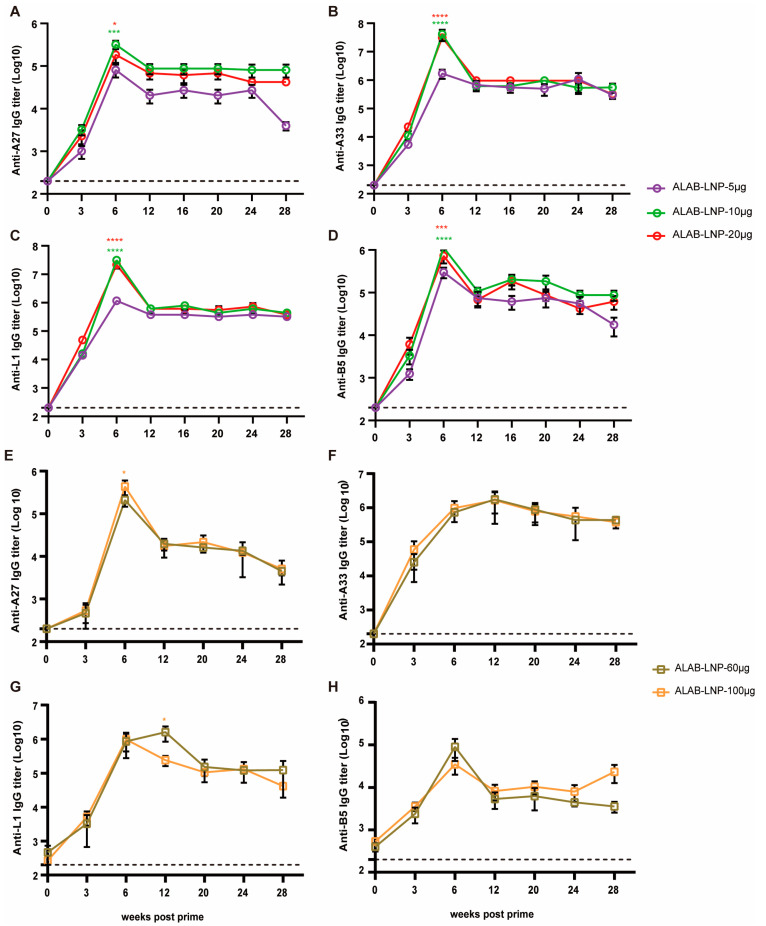
ALAB-LNP induced long-lasting binding antibody response against A27, A33, L1, and B5 proteins in mice and rats. (**A**–**D**) Specific IgG binding titers in sera from ALAB-LNP immunized mice against A27 (**A**), A33 (**B**), L1 (**C**), and B5 (**D**) proteins at different time points post-prime; (**E**–**H**) Specific IgG binding titers in sera from ALAB-LNP immunized rats against A27 (**E**), A33 (**F**), L1 (**G**) and B5 (**H**) at different time points post-prime; Summary data are shown as means  ±  SEM; The dashed line indicates the limit of detection (LOD); Statistical significance was assessed by two-way ANOVA with Tukey’s multiple comparisons test (**A**–**D**) or Sidak’s multiple comparisons test (**E**–**H**); *p* values were indicated by * (*p* < 0.05), *** (*p* < 0.001), or **** (*p* < 0.0001).

**Figure 7 vaccines-12-00385-f007:**
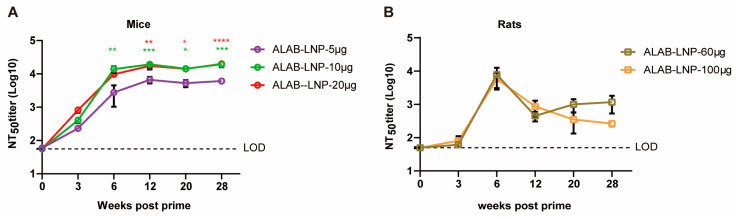
ALAB-LNP induced a long-lasting neutralizing antibody response against VACV in mice and rats. (**A**) Serum neutralizing activity of ALAB-LNP immunized mice against VACV at different time points post-prime; (**B**) Serum neutralizing activity of ALAB-LNP immunized rats against VACV at different time points post-prime; Summary data are shown as means  ±  SEM; The dashed line indicates the limit of detection (LOD); Statistical significance was assessed by two-way ANOVA with Tukey’s multiple comparisons test (**A**) or Sidak’s multiple comparisons test (**B**); *p* values were indicated by * (*p* < 0.05), ** (*p* < 0.01), *** (*p* < 0.001), or **** (*p* < 0.0001).

**Figure 8 vaccines-12-00385-f008:**
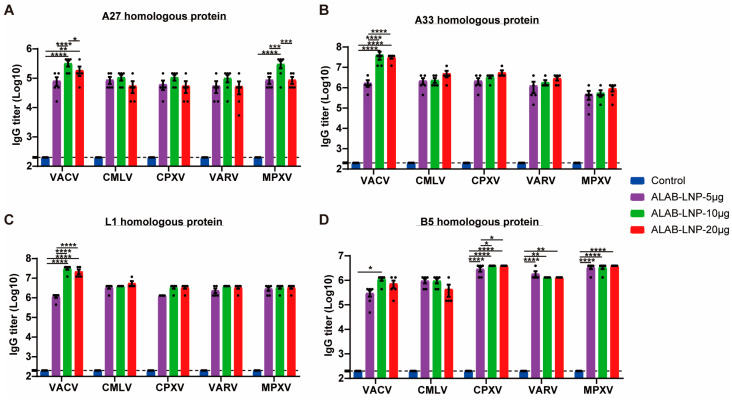
Cross-binding activity against homologous proteins of A27, A33, L1 and B5 of VACV, CMLV, CPXV, VARV, or MPXV, respectively. (**A**–**D**) Binding antibody titers in sera from ALAB-LNP immunized mice against homologous A27 (**A**), A33 (**B**), L1 (**C**), and B5 (**D**) proteins from VACV, CMLV, CPXV, VARV and MPXV, respectively, measured by ELISA; Summary data are shown as means  ±  SEM; The dashed line indicates the limit of detection (LOD); Statistical significance was assessed by two-way ANOVA with Tukey’s multiple comparisons test; *p* values were indicated by * (*p* < 0.05), ** (*p* < 0.01), *** (*p* < 0.001), or **** (*p* < 0.0001).

**Figure 9 vaccines-12-00385-f009:**
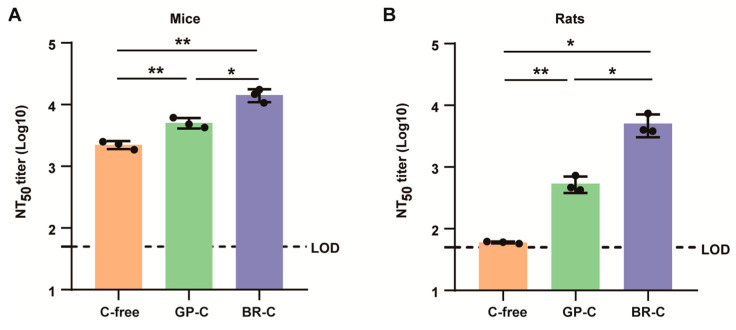
Neutralizing activity was complement-dependent in mice and rats. (**A**) Complement enhanced neutralizing activity in sera from mice; (**B**) Complement enhanced neutralizing activity in sera from rats; Summary data are shown as means ± SEM; The dashed line indicates the limit of detection (LOD); Statistical significance was assessed by unpaired *t*-test; *p* values were indicated by * (*p* < 0.05), ** (*p* < 0.01).

**Figure 10 vaccines-12-00385-f010:**
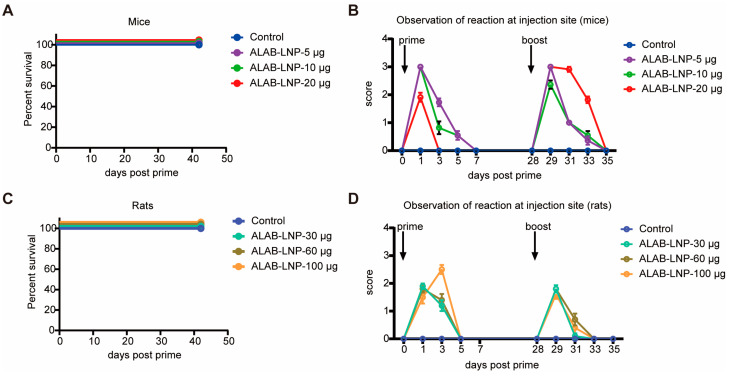
Safety evaluation of ALAB-LNP vaccine candidate in mouse and rat model. (**A**) Survival curve of immunized mice; (**B**) Assessment of side reaction in mice after immunization; (**C**) Survival curve of immunized rats; (**D**) Assessment of side reaction in rats after immunization. The side reactions of injection site were scored as follows: 1, dropsy; 2, swelling; 3, stiffness; 4, immobility.

**Table 1 vaccines-12-00385-t001:** Characterization of ALAB-LNP and Empty-LNP.

Sample ID	EE (%)	Size (nm)	PDI
Empty-LNP	-	76	0.07
ALAB-LNP	90	80	0.05
4Sin-LNP	89	83	0.06

Notes: EE: encapsulation efficiency; PDI: polymer dispersity index.

## Data Availability

The data presented in this study are available on request from the corresponding author.

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
