# Peer review of "A Quadrivalent mRNA Immunization Elicits Potent Immune Responses against Multiple Orthopoxviral Antigens and Neutralization of Monkeypox Virus in Rodent Models"

_vaccines, 2024, doi:10.3390/vaccines12040385_

Round 1

Reviewer 1 Report

Comments and Suggestions for Authors

The authors designed a new mRNA vaccine against Mpox and verified its efficacy and safety in animal studies. Adverse reactions have not yet been eliminated with existing third-generation vaccines, and improved safety is expected. On the other hand, efficacy is generally verified by changes in neutralizing antibodies and by animal experiments to confirm prophylactic efficacy against wild strains. In this regard, the present paper is commendable in following the evaluation method of existing vaccines with the increase in neutralizing antibodies. On the other hand, we believe that the following points require confirmation and verification.

L19-21. There may be no clear factual evidence that smallpox is at increased risk of accidental takeout at this time; the WHO lists the threats posed by smallpox as "global immunity to smallpox, the HIV/AIDS pandemic, and the spread of other immunosuppressive diseases," and does not mention the risk of use for bioterrorism.

See Monitoring variola virus research paragraph 5.

L37-39. Provide a factual basis for the article.

https://jamanetwork.com/journals/jama/fullarticle/190320

L56-58. Provide factual basis WHO lists smallpox threats as "global decline in immunity to smallpox, HIV/AIDS pandemic, and spread of other immunosuppressive diseases" and does not mention the risk of use in bioterrorism. In an earlier review in Viruses, Geoffrey L. Smith et al. describe the lack of information on diversion to bioweapons.

L530-531. Although the authors state that there are no apparent safety issues following mRNA vaccination, there is no description of what criteria were used to evaluate safety in the method.

Reviewer 2 Report

Comments and Suggestions for Authors

This study reports the development of an effective and safe mRNA vaccine candidate against orthopoxviruses. At present, the available orthopox vaccines include vaccinia virus and variously attenuated vaccinia viruses. Although, these vaccines have been proven effective against orthopoxviruses, their use can cause side effects, especially in people with immunodeficiency. This study may be a step towards creating safe vaccines.

The data is solid, but there are a few points that need attention to improve the manuscript's quality.

1. I do not feel the current title captures the essence of the article in the right way. What does “a step closer to a broad poxvirus vaccinemean? Do the authors assume that the vaccine will be effective against yatapox, parapox, and molluscipox viruses pathogenic for humans? This study reports the development of a vaccine candidate against orthopoxviruses. Further down the text, there is also an erroneous text “development of a poxvirus vaccine”. In addition, there is now a “broad orthopoxvirus vaccine” that is a vaccinia virus. This study reports the development of a safe vaccine candidate against orthopoxviruses.

2. I believe that the authors should rewrite the Introduction, due to the large number of errors and information not confirmed by references.

Lines 33-35. Orthopoxviruses belong to the family Poxviridae and the genus orthopoxvirus contains species including smallpox (VARV), monkeypox (MPXV), vaccinia (VACV), camelpox (CMLV) and cowpox virus (CPXV).

Current ICTV Taxonomy Release:

Orthopoxvirus

Species:

Abatino macacapox virus

Species:

Akhmeta virus

Species:

Camelpox virus

Species:

Cowpox virus

Species:

Ectromelia virus

Species:

Monkeypox virus

Species:

Raccoonpox virus

Species:

Skunkpox virus

Species:

Taterapox virus

Species:

Vaccinia virus

Species:

Variola virus

Species:

Volepox virus

Lines 48-59. In this paragraph, the authors describe orthopoxviruses that are pathogenic to humans. It is unclear why they are paying so much attention to the camelpox virus. There is only one article proving its pathogenicity to humans, and the authors do not cite it. Many authors question its pathogenicity, as well as the Abatino virus. At the same time, the authors do not mention, for example, the Akhmeta virus, the pathogenicity of which is widely known.

 Lines 61-64. The genome of vaccinia virus [10], variola virus [11], monkeypox virus [12], camelpox virus [13], ectromelia virus [14] and cowpox virus [15] have been sequenced and the results demonstrated that all viruses are morphologically indistinguishable and antigenically related (The references is missing).

How can genome sequencing demonstrate that viruses are indistinguishable and antigenically related?

Lines 65-68. Using a poxvirus specific tool, accurate gene sets for viruses with completely sequenced genomes in orthopoxvirus were predicted, such that, in all existing orthopoxvirus species, no individual species has acquired protein-coding genes unique to that species.

It is necessary to rewrite the sentence. It is difficult to understand its meaning.

Lines 68-69. This was the foundation for cowpox virus … being effective vaccines against smallpox (The references is missing).

Is the cowpox virus used as a vaccine?

 Line 97. Change Japen to Japan.

 Line 97. Although, these vaccines have been proved efficacious against smallpox in clinical trials.

No clinical trials of smallpox vaccines efficacy have been conducted. Now it is impossible to infect test subjects with a smallpox virus according to international laws.

Lines 97-100. Although, these vaccines have been proved efficacious against smallpox … the vaccines face problems such as side effects, weak immunogenicity and low productivity to meet market need. (The references is missing).

The authors should provide information on the low immunogenicity of these vaccines. I doubt it.

Lines 534-538. future pandemic that possibly caused by zoonotic poxviruses such as monkeypox, cowpox and camelpox, given that the recent monkeypox epidemic raised the poxvirus public health and safety concern and that cowpox and camelpox have been reported infectious to human.

I believe that a pandemic caused by cowpox and camelpox viruses is impossible.    

3. The results obtained by the authors are reliable. The question arises - why did the authors not test the protectivity of the developed vaccine in mice using lethal doses of the mousepox virus (ectromelia)? If the survival rate of animals increases, this may indicate the effectiveness of the developed vaccine.

Reviewer 3 Report

Comments and Suggestions for Authors

Thank you for sharing the manuscript investigating quadrivalent mRNA immunisation eliciting potent immune responses against multiple antigens of orthopoxviruses. The following comments may help to improve the article:

L37-40/499-504/530-535: Lengthy sentences, please shorten. 

L41: Are you referring here to the African or the global outbreak?

L123/L125: Why did you use an ATCC reference strain for vaccinia virus and a patient strain for the monkeypox virus?

L128: How old were the rats? Also 6-8 weeks and also females?

L131: Where within the manuscript can an overview of the designed groups be found? Were the groups same for the mice and the rats? How were the animals immunised? 

L133: Which time point(s) were chosen to collect sera?

L177/190/207/238: Which controls did you run with the assays?

L262/265: Statement about p-value seems duplicated. 

L301-306: A reader would expect to see such information in the material and methods section, but not among the results. Please revise. 

Figure 2: A reader would expect to see Figure 2A in the materials and methods section, where this information is clearly missing as stated before. 

Comments on the Quality of English Language

Some sentences appear pretty lengthy and should be revised for better readability. Also, please go once more in particular over the introduction that seem to contain some rather incomplete sentences that impact its readability.

Round 2

Reviewer 1 Report

Comments and Suggestions for Authors

The authors have made appropriate corrections to the points raised and conveyed accurate factual information.

Reviewer 2 Report

Comments and Suggestions for Authors

The authors have done a lot of work to correct the manuscript. The authors took into account all the recommendations in the revised version of the article In general, the manuscript is well written, the information presented is well organized. I believe that this manuscript can be published in the Vaccines

Reviewer 3 Report

Comments and Suggestions for Authors

Thank you for sharing the revised manuscript. All my comments were addressed sufficiently.